# Microbial Colorants Production in Stirred-Tank Bioreactor and Their Incorporation in an Alternative Food Packaging Biomaterial

**DOI:** 10.3390/jof6040264

**Published:** 2020-11-02

**Authors:** Fernanda de Oliveira, Caio de Azevedo Lima, André Moreni Lopes, Daniela de Araújo Viana Marques, Janice Izabel Druzian, Adalberto Pessoa Júnior, Valéria Carvalho Santos-Ebinuma

**Affiliations:** 1Department of Engineering Bioprocess and Biotechnology, School of Pharmaceutical Sciences, Universidade Estadual Paulista—UNESP, Araraquara 14800-903, Brazil; fernanda.oliveira1@unesp.br (F.d.O.); caio.a.lima@unesp.br (C.d.A.L.); 2Faculty of Pharmaceutical Sciences, University of Campinas—FCF/UNICAMP, Campinas 13083-859, Brazil; amorenilopes@gmail.com; 3Laboratory of Biotechnology Applied to Infectious and Parasitic Diseases, Biological Science Institute, University of Pernambuco-ICB/UPE, Recife 50100-130, Brazil; daniela_viana@yahoo.com.br; 4Department of Bromatological Analysis, Faculty of Pharmacy, Postgraduate Program in Science of Food, Federal University of Bahia, Salvador 40170-115, Brazil; janicedruzian@hotmail.com; 5Department of Biochemical and Pharmaceutical Technology, University of São Paulo, São Paulo 05508-000, Brazil; pessoajr@usp.br

**Keywords:** natural colorants, filamentous fungi, stirred-tank bioreactor, biodegradable films, antioxidant, food package

## Abstract

Natural colorants from microbial fermentation have gained significant attention in the market to replace the synthetic ones. *Talaromyces* spp. produce yellow-orange-red colorants, appearing as a potential microorganism to be used for this purpose. In this work, the production of natural colorants by *T. amestolkiae* in a stirred-tank bioreactor is studied, followed by its application as additives in bio-based films. The effect of the pH-shift control strategy from 4.5 to 8.0 after 96 h of cultivation is evaluated at 500 rpm, resulting in an improvement of natural colorant production, with this increase being more significant for the orange and red ones, both close to 4-fold. Next, the fermented broth containing the colorants is applied to the preparation of cassava starch-based films in order to incorporate functional activity in biodegradable films for food packaging. The presence of fermented broth did not affect the water activity and total solids of biodegradable films as compared with the standard one. In the end, the films are used to pack butter samples (for 45 days) showing excellent results regarding antioxidant activity. It is demonstrated that the presence of natural colorants is obtained by a biotechnology process, which can provide protection against oxidative action, as well as be a functional food additive in food packing biomaterials.

## 1. Introduction

Over the years, humans have used colorants to enhance or restore the appearance of products, especially in the food industry [1,2]. Nevertheless, most of the synthetic colorants used in food products are derived from petroleum and may have harmful effects on human health and the environment [3,4]. Therefore, there is a growing preference for natural additives in foods. Regarding the natural color market, a survey by the Grand View Research, Inc. estimated the world color market for 2025 to be USD 37.49 billion. Specifically, for the food market, the natural colorant consumption will register a 5.9% compound annual growth rate (CAGR) in terms of revenue, in relation to the period from 2018 to 2025 [5].

There are several biological sources that produce natural colorants, such as insects, vegetables, and microorganisms [2,6]. Colorants from microbial origin, especially fungal ones, are of great interest mainly because of the recent improvements in biotechnology and bioprocessing technologies as well as a full control of the production conditions [2,7,8]. Fungus of the genus *Monascus* stands out for presenting good levels of colorant production. However, *Monascus* species co-produce citrinin, a mycotoxin [9]. Thus, other strains such as *Aspergillus*, *Trichoderma*, *Fusarium*, and *Talaromyces* (formerly *Penicillium* spp.) have been the subject of studies in order to produce colorants [8,10,11,12]. In this context, *Talaromyces amestolkiae* (former *Penicillium purpurogenum*) has aroused interest in the production potential of stable and non-toxic yellow, orange, and red colorants [10,13,14,15].

One of the main challenges of using filamentous fungi, such as *Talaromyces,* for metabolite production is the control of microorganism growth and form besides the mass transfer inside the bioreactor [16]. Hyphal concentration greatly influences the fermentation broth viscosity [15]. Meanwhile, the oxygen transfer in submerged cultures is an important factor for efficient fungal growth and an important parameter for a high yield of fungi colorants [17]. At a high cell concentration, the culture medium viscosity increases and it limits the oxygen supply for the cells [4,18]. Increasing the stirring speed can maintain dissolved oxygen concentration at high levels, however, potential damage to the fungal hyphae can limit the impeller speed and, consequently, the oxygen and nutrient transfer capability of a bioreactor [19,20]. In order to control cell growth and colorant biosynthesis, the well-directed process parameter shift represents a valuable control strategy [9,21]. In the production of *Monascus* colorants, the pH-shift strategy has been proven to be efficient in colorants biosynthesis [17,22]. Hence, great efforts to improve the bioreactor cultivation conditions are necessary for enhancing the large-scale production of *Talaromyces* colorants.

Biosynthetically, many fungi colorants are polyketides. Polyketide colorants range in structure from tetraketides to octaketides, which have four or eight C_2_ units that contribute to the polyketide chain. The structure of the polyketides does not exhibit localized negatively-charged ions. These molecules often have a polyunsaturated function, i.e., a ring system, one or more carbonyl groups, carbolic acid, and functional ester or amide groups that absorb in the UV-visible spectrum [23]. One class of polyketide colorants are azaphilones, which are hexaketide colorants with pyrone-quinone structures and a chiral quaternary center [24]. Investigations focus on six major azaphilone colorants: Rubropunctamine and Monascorubramine (red); Rubropunctatin and Monascorubrin (orange); Monascin and Ankaflavin (yellow) [25]. One of the most interesting characteristics of azaphilone colorants is their potential use as a functional food colorant, because they may exert antioxidant and antimicrobial activities, relatively low cytotoxicity, as well as immunosuppressive, antiviral, anticancer, and cholesterol-reducing properties [26]. Moreover, our research group previously reported that *T. amestolkiae*-fermented broth presented low cytotoxicity against fibroblast cells and effective antimicrobial activity against *Staphylococcus aureus*, a representative food contaminant [13]. In this sense, bio-based materials with antioxidant properties are gaining attention in the food sector [27], mainly because non-ecofriendly packages are considered an environmental problem [28].

The application of natural compounds in active packaging has been increasingly applied in order to produce materials that interact with packaged foods, positively modifying their sensory and nutritional properties, preventing the deterioration of food, mainly caused by lipid oxidation and microbial growth [29]. Particularly, foods with high-fat content and especially those with a high degree of unsaturation are susceptible to deterioration by oxidation. In this way, alternative packaging technologies based on the inclusion of antioxidants compounds can create a barrier, improving the stability of packaged products sensitive to oxidation [30]. Therefore, the use of the *T. amestolkiae* natural colorants as an added component in bio-based films represents an alternative for increasing the properties of food packaging.

This work addresses the production of natural colorants by *T. amestolkiae* in a stirred-tank bioreactor. The effects of stirring speed and pH-shift as a control strategy on colorant accumulation are evaluated. After the selection of the best operational condition, the fermented broth containing the colorant is incorporated into cassava starch biodegradable films. The bio-based films prepared are characterized with respect to thickness, water activity, and total solid content. In the end, the cassava starch-based films are used to pack butter, supporting the applicability of natural colorants as additives in bio-based films. The effectiveness of the *T. amestolkiae* colorants protection against oxidative action is addressed on the basis of the peroxide content under accelerated oxidative conditions.

## 2. Materials and Methods

### 2.1. Materials

Sucrose and yeast extract were purchased from Synth (São Paulo, Brazil) and Acumedia (Lansing, MI, USA), respectively. Cassava starch (composed by 23.5% amylose and 64.2% amylopectin) was donated by Cargill Agrícola S.A. (Porto Ferreira, SP, Brazil). Commercial butter was obtained from Imperial (Bahia, Brazil). Low-density polyethylene (LDPE) film (0.020 mm thickness and 15.86 × 10^−8^ g H_2_O·mm/m^2^·h·kPa water vapor permeability) was purchased from local markets (Salvador, BA, Brazil). All of the other reagents were of analytical grade.

### 2.2. Microorganism Maintenance and Colorant Production

*Talaromyces amestolkiae* DPUA 1275 was generously provided by the Culture Collection of the Federal University of Amazonas (DPUA, Manaus, AM, Brazil). The cultures preserved in distilled water were reactivated in Czapeck Yeast Extract Agar (CYA) and maintained at 30 °C for 7 days. The CYA medium had the following composition (g/L in deionized water): K_2_HPO_4_ (1.0), yeast extract (5.0), sucrose (30.0), agar (15.0), and 10 mL/L of concentrated Czapeck [(g/100 mL of deionized water): NaNO_3_ (30.0), KCl (5.0), MgSO_4_·7H_2_O (5.0), and FeSO_4_·7H_2_O (0.1)] [10].

The production process was composed of three phases: pre-inoculum, inoculum, and submerged culture. For the pre-inoculum preparation, a loop of fungus from stock culture was inoculated on a CYA plate and maintained in the same reactivation conditions. For inoculum, five mycelial agar discs (8 mm diameter) of *T. amestolkiae* were punched out from the pre-inoculum with a self-designed cutter and transferred to 50 mL of submerged culture medium in 250-mL Erlenmeyer flasks incubated at 30 °C and 150 rpm for 72 h. The inoculum medium was CYA liquid (without the addition of agar). Then, the entire volume obtained (0.2 L) was aseptically transferred to a single flask and then transferred to the bioreactor. The composition of the submerged culture medium was similar to that used for inoculum preparation, except for the concentration of sucrose and yeast extract, which were 48.50 and 11.80 g/L, respectively [10]. Both the inoculum medium and the culture broth had their pH adjusted to 4.5 with HCl (5 M) and were autoclaved at 121 °C for 15 min. The submerged culture was performed in a stirred-tank bioreactor Bioflo^®^ 115 (New Brunswick, Edison, NJ, USA) with 3 L of working volume equipped with two Rushton impellers submersed into the bulk liquid. After sterilization, the bioreactor containing 1.8 L of culture medium was supplied with 0.2 L of the inoculum. The following operational conditions were kept constant by the bioreactor controllers: temperature at 30 °C and aeration rate at 2.0 vvm. The experiments were carried out for 240 h, and every 24 h, an aliquot was withdrawn. At the end of the bioprocess, the fermented broth was filtered first using filter paper Whatman #1 (Whatman, Marlborough, England) and later using a 0.45 μm filter acquired from Millipore. The filtrated samples were used to determine the production of yellow, orange, and red colorants as well as sucrose consumption. In fact, during the cultivation of *Talaromyces* spp., the three natural colorants (yellow, orange, and red) were produced at the same time. In order to guarantee that the colorants were produced, we analyzed the specific wavelength for each one of them, according to Section 2.3. 

In the first set of experiments, the effects of the stirring speed from 100 to 600 rpm were evaluated. Next, pH-shift experiments were carried out at 500 rpm, adjusting the pH to 8.0 with NaOH (1 M) after the sucrose depletion (96 h of cultivation).

### 2.3. Analytical Methods

The sucrose concentration was determined according to the methodology described by Dubois et al. [31] and the pH was measured using a pH-meter (model MPC 227, Mettler Toledo, Columbus, OH, USA). The production of extracellular colorants was estimated by spectrophotometric analysis (model DU 640, Beckman, Irving, TX, USA) by reading the absorbance of supernatant at 410, 470, and 490 nm, which corresponds to the maximum absorbance for yellow, orange, and red colorants, respectively. The results were expressed in terms of absorbance units (AU) [10].

### 2.4. Kinetic Parameters

To calculate the substrate conversion factor (sucrose) in product (*Y_P/S_*) and productivity (*P*), we employed Equations (1) and (2), respectively:(1)$$Y_{P/S}=\;\frac{Abs_{f}-\;Abs_{0}}{S_{0}-\;S_{f}}$$
(2)$$P=\;\frac{Abs_{f}}{t}$$
where: *S*_0_ and *S_f_* are the initial and final sucrose concentrations; *Abs_f_* and *Abs*_0_ are the final and initial colorant absorbance, respectively, and *t* is the time (h).

### 2.5. Preparation of Cassava Starch-Based Films and Incorporation of Natural Colorants

The cassava starch-based films were prepared according to the methodology described by Silva et al. [32]. In this way, the bio-based films were prepared by the casting technique, mixing cassava starch (4.0 wt%), plasticizers (0.7 wt% sucrose and 1.4 wt% inverted sugar), and the fermented broth containing the colorants (5.0 wt%). The fermented broth was used because of the synergy between the colorants molecules and the different metabolites present on fermented broth that increase the antioxidant power. Dispersions were heated (70 ± 2 °C), degassed for 30 min in ultrasonic bath to remove the bubbles, placed in polystyrene Petri dishes (150 × 15 mm), and dehydrated in an oven with airflow and circulation at 35 ± 2 °C for 24 h. A cassava starch-based film without addition of the fermented broth was prepared and used as control. Resulting bio-based films were stored in desiccators (at 23 ± 2 °C with 60 ± 2% relativity humidity) with a supersaturated solution of magnesium nitrate for 48 h before the characterization studies.

### 2.6. Cassava Starch-Based Films Characterization

The thickness was evaluated using a flat parallel surface micrometer (Mitutoyo model 103–137, precision 0.002 mm). Six measurements were taken at random positions around the film sample. The results were expressed as the mean ± respective standard deviations and used for the calculation of the contact area (mm).

The water activity (aw) was measured with a Decagon, Aqualab Lite, as calibration standards used pure water (aw of 1.000 ± 0.001%) and LiCl (aw of 0.500 ± 0.015%). Preconditioned samples (4 cm^2^) were cut from the center of the films and evaluated in triplicate at room temperature (25 ± 2 °C).

The total solid content was determined by measuring the weight loss of films upon drying (105 °C) until constant weight. This determination was performed in triplicate. The results were expressed as the mean ± respective standard deviations.

### 2.7. Application of Cassava Starch-Based Films for Packaging Butter

Butter was packed in the cassava starch-based films with colorant additives. Square-shaped films (5 × 2 cm) of 0.164 and 0.212 mm in thickness were molded (Sealer Sulpack SM 400 TE, Caxias do Sul, Rio Grande do Sul, Brazil). Butter samples were homogenized and frozen in small pieces (with 3 × 2 cm and 10.00 ± 0.54 g), after they were involved with the film, the bubbles of oxygen were removed, and the film was sealed.

The samples of packaged butter were stored for different periods of time (0, 7, 15, 30, and 45 days) and under accelerated oxidative conditions (64% of relative humidity at 25 ± 2 °C). These analyses were carried out in a dark room to avoid the effects of light interference (i.e., color degradation) in the samples. For this, three types of packages were prepared: cassava starch-based films with colorant additives (CFC), cassava starch-based films without colorant additives (CF), and conventional plastic (CP). In addition, unpackaged butter (U) was used as control. To evaluate the antioxidant action of the films formulation, the oxidative stability of the butter stored under accelerated oxidative conditions was monitored for 45 days and the peroxide content (PC) was determined by titration method according to the methodology described by the Association of Official Analytical Chemists–AOAC (2000) [33].

### 2.8. Statistical Analysis

The data were analyzed by ANOVA using a StatSoft v.7 program (StatSoft, Inc., Tulsa, OK, USA). The Tukey test was used to evaluate the mean difference in results at the 95% level of significance.

## 3. Results and Discussion

### 3.1. Production of Natural Colorants in Bioreactor: Effect of Stirring Speed

The high viscosity of the medium can result in oxygen diffusion limitation, due to the combination of the high biomass concentration and the fungal morphology. Hence, efforts have been made in order to overcome mass transfer problems [16,34]. Therefore, the effects of the stirring speed and pH were investigated by our group. In this sense, the production of natural colorants by *T. amestolkiae* in a stirred-tank bioreactor was evaluated, we maintained the aeration rate constant at 2.0 vvm and varied the stirring speed from 100 to 600 rpm in order to achieve proper oxygen diffusion in our cultivations. The results are depicted in Figure 1.

As can be seen in Figure 1, the stirring speed influenced the colorants’ yield. At 100 rpm, the colorants’ production stopped after 120 h, probably because under this condition the microorganism distribution inside the bioreactor was not homogeneous. It is usually observed that submerged cultures of filamentous fungi present a pseudo plastic behavior (i.e., a non-Newtonian characteristic) [35], and low speed, associated with high medium viscosity, does not allow for system homogeneity. The main role of stirring in a bioreactor is to improve heat and mass transfer to achieve homogeneity in the system. A proper mixing of components in the culture medium is necessary to ensure an adequate flow in submerged aerobic culture [36]. The stirring of the culture medium may also cause different effects in filamentous microorganisms. Among them may be included: cell wall disruption, changes in filamentous morphology, variation in growth efficiency and growth rate, and the variation in the formation rate of the desired bioproduct [18,36]. The stirring process may damage both mycelial pellets and hyphae structures. The fragmentation of hyphae can result in the creation of small parts of these structures, promoting a new center for biomass growth, or may cause damage to the hypha allowing release of the cytoplasm. On the other hand, total pellet breakdown may occur, and, probably in this case, total autolysis or aggregates can occur [37].

Increasing the stirring speed could maintain dissolved oxygen concentration at high levels being a key parameter for high yield of *T. amestolkiae* colorants in the stirred-tank bioreactor cultivation. As the stirring speed increased, the homogeneity of the culture medium could be achieved, and it was possible to determine the natural colorant production. Since the oxygen transfer is directly proportional to the shear force and it is strictly related to the morphology of the filamentous fungus, stirring speeds above 600 rpm were not evaluated in this study. Considering the production of orange and red colorants, there was no statistical difference between the results achieved at 500 rpm and 600 rpm (statistical test: one-way ANOVA, *p* = 0.0994). However, as 500 rpm promote a lower shear stress, this stirring speed was chosen for the next experiments. In this condition, the absorbance of the yellow, orange, and red colorants was 1.71 AU_400nm_, 0.71 AU_470nm_, and 0.69 AU_490nm_, respectively. Comparing the colorant production achieved with results previously reported by our research group [10], using the same culture media but in orbital shaker incubator, there was a decrease of 5-fold for the red colorant production in the bioreactor. This result confirms the statement that the scale-up from orbital shaker incubator to large fermenters is a difficult task. In fact, several conditions of cultivation can change in the bioreactor, mainly the oxygen transfer rate in the medium.

Figure 2 shows the results of sucrose consumption and pH over time of submerged culture of *T. amestolkiae* in bioreactor varying the stirring speed. All values are detailed in Appendix A in the supplementary material.

Regardless of the stirring speed evaluated, the sucrose consumption showed a similar profile (Figure 2A). In general, the concentration of sucrose decreased during the cultivation and a pronounced consumption of sucrose at 96 h occurred. At the end of cultivations, the final sucrose concentration was close to 2 g/L in all conditions studied.

With respect to pH variation, regardless of the stirring speed evaluated, there was an increase in its value throughout the cultivation, with the pH value of 4.5 reaching values close to 7.5 after 240 h of cultivation (Figure 2B). This profile was similar to what occurred in the submerged culture of *T. amestolkiae* (former *Penicillium purpurogenum*) in the orbital shaker incubator studied by Santos-Ebinuma et al. [10]. Therefore, pH 4.5 is suitable to provide good cell growth.

### 3.2. Production of Natural Colorants by Submerged Culture in Bioreactor: pH-Shift Strategy

The stirring speed of 500 rpm was selected to evaluate the pH-shift control strategy on *T. amestolkiae* colorant accumulation. At 500 rpm, the initial pH of 4.5 changed to approximately 8.0 at the end of cultivation (Figure 2B). According to Orozco and Kilikian (2008) [38], the production of red colorants by the submerged culture of *Monascus purpureus* in a bioreactor can be favored by a pH change. These authors selected a pH of 5.5 for the growth step and pH 8.0 for the production step. Thus, a similar strategy was used in this work in order to increase the colorant production in the bioreactor cultivation. In this way, at 500 rpm of stirring speed, a cultivation was carried out and, after 96 h, the pH was changed to 8.0 by the addition of 5 M NaOH. The time of 96 h was selected according to the experiments carried out that varied the stirring speed, because it was observed that, at this point, there was a marked decrease in the concentration of the primary carbon source (Figure 2B). Figure 3 depicts the production of yellow, orange, and red colorants, as well as the pH for the assays performed at 2.0 vvm, 30 °C, and 500 rpm.

After the pH-shift, it is possible to observe a significant increase in the production of all the colorants, with this increase being more significant for the orange and red ones, both close to 4-fold. Furthermore, the influence of the pH-shift on colorant production was evident. The maximum yield of yellow, orange, and red colorants was 3.20 AU_400nm_, 2.56 AU_470nm_, and 2.45 AU_490nm_, respectively. These results corroborate those found by Orozco and Kilikian (2008) [38], who mention that different pH levels during the growth and production steps may lead to an improvement in the colorant production. Although the colorant production with the pH-shift strategy was greater than for the first set of experiments, the product yield was still lower than for the best condition in the orbital shaker incubator, which shows that further studies are necessary to improve the production of *T. amestolkiae* natural colorants in a stirred-tank bioreactor.

Compared with no pH change condition, all kinetic parameters evaluated improved with the pH-shift control strategy (Table 1). The rate of the substrate conversion to product increased from 0.037 to 0.067 AU.L/g for the yellow colorants and from 0.015 to 0.054 and 0.051 AU.L/g for the orange and red colorants, respectively. Similarly, the productivity of all colorants increased in this second experimental condition.

Generally, in bioreactor cultivation, reproducible kinetic growth of filamentous organisms is difficult to obtain. This phenomenon occurs frequently, due to mycelial aggregation in fermentation broths with long hairy mycelial morphologies that can adhere to surfaces and form a growing biofilm [39]. Many fungal strains grow preferentially on surfaces and will develop thick layers on walls and surfaces in a submerged fermentation process [40]. In the same way, reproducible samples to measure apparent viscosity are difficult to obtain, hence, it was also deemed appropriate to omit the time course for biomass and viscosity.

Even though it is desirable to produce fungal colorants comprised of only one single color component [22], it is known that the fermented broth is composed of yellow, orange, and red colorants [13,14,15]. However, the careful selection of pH and nitrogen sources is a valid approach to produce extracellular extracts comprised of one predominant color component [14]. It has been widely accepted by the scientific community that orange colorants are the first biosynthetic product and the other colorants derive from the orange ones [41]. In this way, the extracellular colorant composition and concentration is dependent of the orange colorants produced that are transformed into red and yellow colorants under some specific fermentation conditions. For the *Monascus* genus, a low pH (pH 2.5 and 4.0) provides intracellular extracts composed mainly of orange colorants, independently of the nitrogen source employed. At acidic conditions, the secretion of orange colorants into the broth and the reaction with any amino unit for red colorants formation is limited [22]. Meanwhile, a change of pH to levels closer to neutral modify the extent of transformation of orange colorants into red ones [22]. Additionally, for *T. amestolkiae*, the synergistic effect of a low pH and nitrogen source is mainly important in the production and excretion of red colorants. A previous study of our research group presented the production of a glutamic acid-red colorant complex by *T. amestolkiae* in a chemically defined medium with monosodium glutamate (MSG) as the nitrogen source [14]. In the presence of MSG, deep yellow colorants were derived from neutral and basic pH, while deep red colors were derived from acidic pH. The glutamic acid–colorant complex seems to be more water-soluble than those produced in nitrogen complex media (meat extract and meat peptone) by the same strain [13].

In this work, yeast extract was used as complex nitrogen source and the synergistic effect of this nitrogen source with pH favored the increase of the relation between orange colorants and yellow/red ones. This is a reflection of the overall increased rate of substrate to product and productivity. It seems that a close to neutral pH favored the reduction of orange colorants to yellow ones (relation Abs_O_/Abs_Y_ < 1.0 duplicated with pH-shifting), while the specific red conversion was limited (relation Abs_O_/Abs_R_ > 1.0). Consequently, the extracellular extracts contained mainly yellow colorants. Therefore, extracellular red colorants derivatives were not mainly produced at a high pH. It can be considered that natural colorants can be obtained by the submerged culture of *T. amestolkiae* in a bioreactor, however, the culture conditions can be improved and the production increased by varying mainly the pH during cultivation. In order to study a possible application of the colorants produced, the incorporation of the fermented broth into an alternative food-packaging material based on its oxidative protection effect was evaluated.

### 3.3. Preparation and Characterization of Cassava Starch-Based Films

Biodegradable films were produced by a formulation of cassava starch, plasticizers, and the fermented broth containing the *T. amestolkiae* colorants produced in stirred-tank bioreactor. The incorporation of natural additives from plants for active packaging has been increasingly applied, such as: coffee-cocoa [42,43], carotenoid and yerba mate extract [44], green tea and palm oil extract [45], among others. However, natural additives produced by biotechnology is an innovative approach. The bio-based films produced were characterized in terms of thickness, water activity, and total solids content. The results are depicted in Table 2.

The physical properties of the films were determined to evaluate the influence of the colorant incorporation, since they could modify the structure of the polymer matrix, e.g., by weakening the inter-chain bonds. The incorporation of natural colorants in the film formulation caused a significant reduction of film thickness, around 10%, in relation to films with no additives. Other works also reported a reduction in thickness of films formulated with cassava starch and natural additives [43,46]. The control of the thickness in the biodegradable films is an important step, since variations in this parameter can affect some films properties, such as mechanical and barrier properties, which can compromise the package performance [45]. Thickness was also the only parameter addressed that was affected significantly.

The variations in the total solid content in the films after colorant incorporation were not significant, with values of 88.96 and 89.80%, respectively. In general, the incorporation of natural additives in cassava starch-based films does not affect significantly the solid content [43].

Additionally, no significant difference (<0.05) for water activity (aw) between the films produced was observed. These results demonstrate that the presence of natural colorants did not change the water activity of the films. Water activity brings information about the amount of free (also referred as unbound or active) water present in a sample [47] being a parameter used in food preservation. Low water activity reduces the availability of water to microorganisms, avoiding undesirable chemical changes for the storage of products [32]. As water migrates from areas of high aw to areas of low aw, it is important to have low values of aw (<0.600 according to Mathlouthi, 2001 [48]) for food product design. As the aw was around 0.6 for both films, they can be considered as food product packages. In this way, the cassava starch-based films were used to pack butter samples.

### 3.4. Oxidative Stability of the Packaged Butter during Storage

The oxidative stability of the butter packaged in different formulations, namely: cassava starch-based films with colorant additives (CFC), cassava starch-based films without colorant additives (CF), and conventional plastic (CP) was evaluated. Unpacked butter (U) was used as the control. To this purpose, the packed or unpackaged butter was stored for 45 days under accelerated oxidative conditions (64% of relative humidity and 25 °C) and monitored through peroxide content (PC) at 7, 15, 30, and 45 days of storage. Appendix A from Supplementary Material shows an image of the butter packaged in each formulation evaluated. Figure 4 shows the results of PC for each condition over the 45 days of storage. All values are detailed in Appendix A in Supplementary Material.

From Figure 4 can be seen that, as expected, the PC of the U was higher than of the packaged one. The CF and CP showed higher peroxide indices than CFC. This demonstrates that natural colorants act as both a photooxidative protector and antioxidant agent, which allow a higher oxidative protection of the packaged product when compared to the other conditions evaluated.

After 45 days of storage, the PC increased 329.0% for the exposed butter, 239.7% for the CP, and 231.1% for the CF. While for the CFC, the PC increased only 118.5%. Therefore, even without antioxidant additives (CF), cassava starch-based films have a greater oxidative protection effect when compared to conventional package films (CP). However, the formulation with colorants incorporated in cassava starch-based films (CFC) enhanced the oxidative protection of butter, with a reduction in peroxide value. This shows a synergistic action of the bio-based film and the colorants, improving the shelf life of fatty products. In this way, *T. amestolkiae* colorants can be considered as potential natural antioxidants for the stabilization of lipid-containing foods for active packing formulation. However, additional studies on the concentration of colorants in the formulation should be performed in order to avoid a possible pro-oxidant effect. Since a high content of antioxidant additives can act as a pro-oxidant agent [44,45].

## 4. Conclusions

This work evaluated the influence of the stirring speed and pH on the production of natural colorants by *T. amestolkiae* in a stirred-tank bioreactor. The greatest colorant production occurred at 500 rpm, under the pH-shift strategy from 4.5 to 8.0 during the production phase. Although the regulatory mechanisms of the biosynthesis of *T.*
*amestolkiae* colorants are not yet clear, the relationship between the culture conditions and the colorant formation it is of great importance. Despite the difficulties in terms of viscosity of the medium, during the submerged cultivation of filamentous fungi in a bioreactor, there is the possibility of using strategies to control growth and metabolite production, especially in terms of stirring and pH. Moreover, this work demonstrated the potential of natural colorants to be included in biodegradable film formulation. The incorporation of colorants in cassava starch-based films provides oxidative protection in packaged butter, by the decrease of peroxide index. Hence, *T. amestolkiae* colorants have great potential for use as functional food colorants. As future perspectives, our group intends to evaluate the surface morphologies of the films with or without colorants by scanning electron microscope in order to gather supplementary data about the characteristics and stability of our formulations extending, for example, its shelf-life application.

## Figures and Tables

**Figure 1 jof-06-00264-f001:**
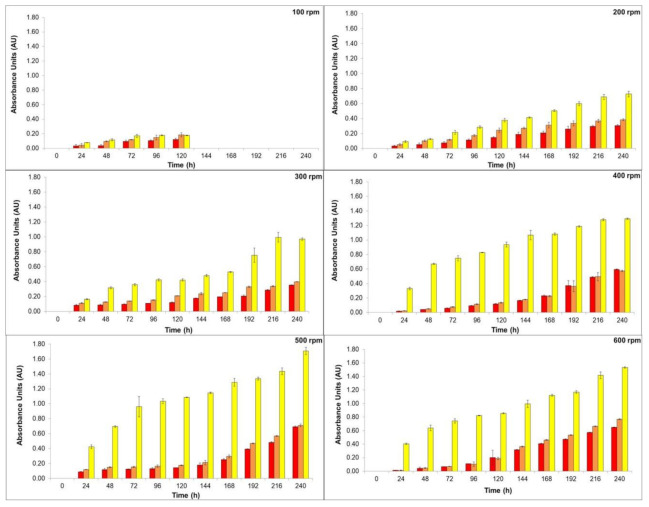
Production of yellow (yellow bars), orange (orange bars), and red colorants (red bars) by submerged culture of *T. amestolkiae* in stirred-tank bioreactor at 2.0 vvm and 30 °C by varying the stirring speed (100 to 600 rpm) as a function of time (0 to 240 h). The error bars represent 95% confidence levels for the mean of three independent assays.

**Figure 2 jof-06-00264-f002:**
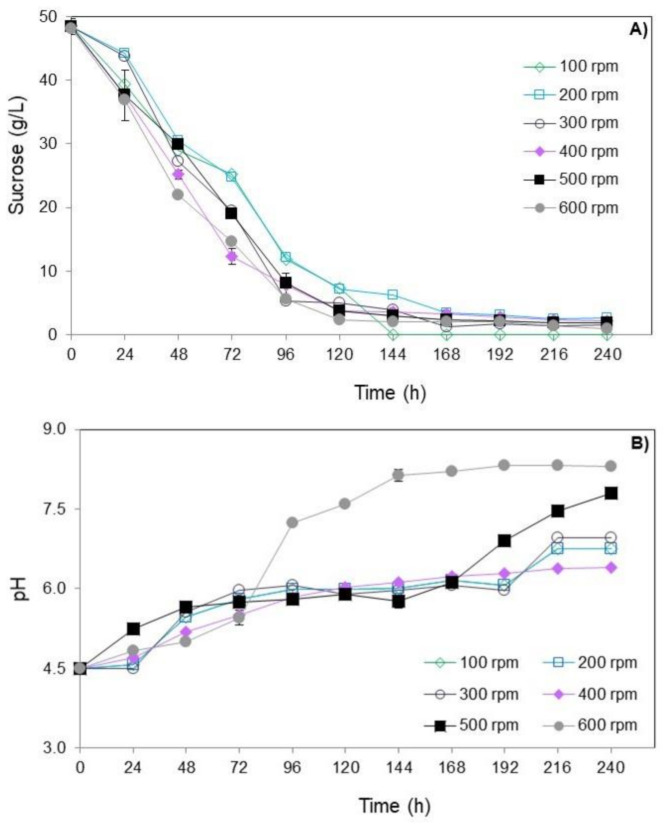
Consumption of sucrose (**A**) and pH (**B**) over time (0 to 240 h) of submerged culture of *T. amestolkiae* in stirred-tank bioreactor varying the stirring speed (100 to 600 rpm) at 2.0 vvm and 30 °C. The error bars represent 95% confidence levels for the measurements.

**Figure 3 jof-06-00264-f003:**
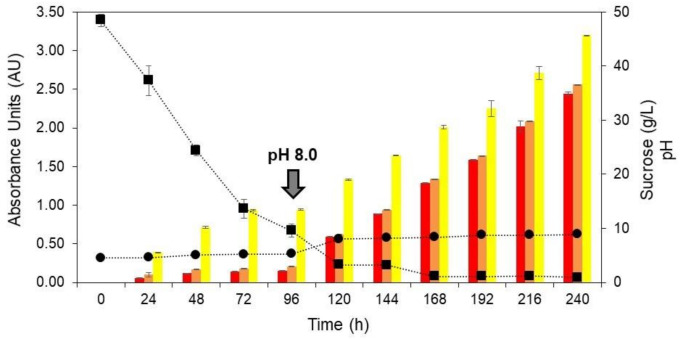
Production of yellow (yellow bars), orange (orange bars), and red colorants (red bars), pH (circle) and sucrose (square) as a function of time (0 to 240 h) during the submerged culture of *T. amestolkiae* in a stirred-tank bioreactor at 2.0 vvm, 30 °C, and 500 rpm. The black arrow indicates the pH-shift from 4.5 to 8.0 at 96 h for the production step. The error bars represent 95% confidence levels for the measurements.

**Figure 4 jof-06-00264-f004:**
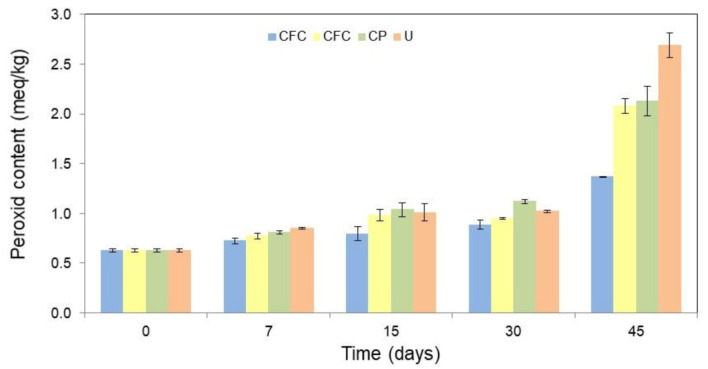
Peroxide content (PC, meq/kg) in the butter as a function of the packaging film used, such as: cassava starch-based films with colorant additives (CFC), cassava starch-based films without colorant additives (CF), conventional plastic (CP), and unpackaged butter (U), during different periods of time (0, 7, 15, 30, and 45 days) in storage. The error bars represent 95% confidence levels for the measurements.

**Table 1 jof-06-00264-t001:** Kinetic parameters calculated for substrate conversion factor (sucrose) in product (Y_P/S_) and productivity during the submerged cultivation of *T. amestolkiae* in a stirred-tank bioreactor for the production of yellow (Y, AU_400nm_), orange (O, AU_470nm_), and red colorants (R, AU_490nm_) without a pH change and after a pH-shift from 4.5 to 8.0 at 96 h of cultivation.

Experimental Condition	Y_P/S_ (AU·L/g)			Productivity (AU/h)			Abs_O_/Abs_Y_	Abs_O_/Abs_R_
	Y	O	R	Y	O	R		
Without pH change	0.037	0.015	0.015	0.007	0.007	0.007	0.415	1.025
pH-shift strategy	0.067	0.054	0.051	0.013	0.011	0.010	0.800	1.049

Abs_O_/Abs_Y_: relationship between the production of orange and yellow colorants; Abs_O_/Abs_R_: relationship between the production of orange and red colorants.

**Table 2 jof-06-00264-t002:** Characterization of biodegradable films in the presence or absence of natural colorants in terms of thickness, water activity (aw), and total solids.

Characterization/Parameters Analyzed	Biodegradable Films Containing Natural Colorants *	Biodegradable Films Without Natural Colorants *
Thickness (mm)	0.147 ± 0.009 ^a^	0.163 ± 0.013 ^b^
aw (%)	0.612 ± 0.041 ^a^	0.604 ± 0.050 ^a^
Total solids	89.80 ± 0.540 ^a^	88.96 ± 0.500 ^a^

* Statistical test: one-way ANOVA (with Welch correction for unequal variances). The results were expressed as the mean ± respective standard deviations. Equal letters in the same line represent equal values to a significance level of 5%.

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
