# Peer review of "Microbial Colorants Production in Stirred-Tank Bioreactor and Their Incorporation in an Alternative Food Packaging Biomaterial"

_jof, 2020, doi:10.3390/jof6040264_

Round 1
Reviewer 1 Report
The manuscript by de Oliveira and colleagues deals with the production of yellow, orange and red natural colorants by T. amestolkiae microbial fermentation in stirred-tank bioreactor. Authors evaluated the effects of different parameters such as pH, stirring speed and sucrose on the final pigment content production. Moreover, authors tried an experimental application using the fermented broth containing the colorants for the preparation of cassava starch-based films, with the aim to incorporate functional activity in biodegradable films for food packaging.
Generally, the manuscript is well written in all its parts. Especially, abstract and introduction treat the current state of art, taking in account the real problem of chemical colorant use in food industry, and the potential and innovative application of those of natural origin. I would just suggest to add some information about the kind of molecules responsible for the yellow, orange and red colors. Are there some chemical characterization?
Materials and Methods are well described. The Results are of interest, and report innovative data. Furthermore, data are adequately discussed. However, the major concerns are related to the statistical analyses and to the figures reported in the manuscript.
Figure 1: Please report the standard deviation in the Figure. If there are graphical problems, consider to insert an additional table in which the raw measurements are reported with the correspective standard deviation. If the authors will decide to insert the supplementary table, they should remember to insert the reference to the new supplementary table in the caption of the figure. Moreover, the authors should perform ANOVA test on the content of the three pigments as a function both of time and stirring speed. Then, they should report different letters on the top of each bar indicating statistical differences. Also in this case, authors should remember to report information regarding statistical analysis in the caption of the figure.
Figure 2: Please, report the letter of the panel on the top-right of each panel, and delete the graph “)”. Moreover, since JoF has no additional cost for colour images, consider to redraw the image using different colours for the different treatments. Finally, the authors should perform ANOVA test on the content of the sucrose and on pH values as a function of time. Considering the problems in reporting the letters in each panel of the graph, I suggest to report them in a new supplementary table. Authors should remember to report in the caption of the figure both information regarding statistical analysis and reference to the new supplementary table.
Figure 5: Please report the standard deviation in the Figure. Moreover, the authors should perform ANOVA test on the peroxide content as a function of time. Then, place different letters on the top of each bar indicating statistical differences. Also in this case, authors should remember to report information about statistics in the caption of the figure.
Moreover, some minor mistakes were found during the reviewing procedure
Affiliation section: immediately after the affiliation, the e-mail and the initials of each author must be reported into the brackets (ex. Department of Engineering Bioprocess and Biotechnology, School of Pharmaceutical Sciences, Universidade Estadual Paulista – UNESP, Araraquara, Brazil; fernanda.oliveira@unesp.br (F.O.); caio.azevedolima@unesp.br (C.A.); etc…)
Line 41: “may have harmful effects on human health and in the environment” -> for the environment
Line 43: “cosmetics” -> Authors give information about the use of colorants in food industry, and not in pharmaceutical industry. Please, remove cosmetic, or add information regarding the use of colorants in pharmaceutical industry.
Line 43: “Regarding the natural color market, a survey by Grand View Research, Inc. estimates” -> estimated
Line 69: “One of the most interesting characteristics of natural fungal colorants is the potential of their use as a functional food colorant” -> is their potential use as functional food colorant.
Line 70: “they can present antioxidant” -> they may exert
Line 74: “a representative of food contaminant”; Staphylococcus aureus -> Staphylococcus aureus
Line 83-84, 112 and 390 “T. amestolkiae” -> T. amestolkiae
Line 106-108 Report subscripts in the different formulas
Line 128 Talaromyces -> Talaromyces
Line 187. The post hoc test is not Turkey, but Tukey
Section 3: please, rename “Results and Discussion”
Line 194 “T. amestolkiae” -> T. amestolkiae
Line 222-223: “The highest colorant production occurred at 500 rpm of stirring” -> this statement does not have sense since ANOVA analysis was not performed. Please see comment on Figure 1.
Figure 3: please remove the halo from the trend lines. Please, see note to the statistical of the Figure 1.
Line 266: “it is possible to observe the significant increase” -> “it is possible observing a significant increase.
Line 278 I suppose that UA is AU. Check also Table 1
Author contribution: the names of the authors must be reported as acronyms (ex. F.O.).
Author Response
Reviewer #1
The manuscript by de Oliveira and colleagues deals with the production of yellow, orange and red natural colorants by T. amestolkiae microbial fermentation in stirred-tank bioreactor. Authors evaluated the effects of different parameters such as pH, stirring speed and sucrose on the final pigment content production. Moreover, authors tried an experimental application using the fermented broth containing the colorants for the preparation of cassava starch-based films, with the aim to incorporate functional activity in biodegradable films for food packaging.
- Generally, the manuscript is well written in all its parts. Especially, abstract and introduction treat the current state of art, taking in account the real problem of chemical colorant use in food industry, and the potential and innovative application of those of natural origin. I would just suggest to add some information about the kind of molecules responsible for the yellow, orange and red colors. Are there some chemical characterization?
RESPONSE: We thank the reviewers for the time put on this manuscript, which made it better written and explained. The issues raised are addressed properly and are highlighted with a different color in the revised manuscript. We introduced a background on chemical characterization of fungi colorants and added more information in the revised manuscript. In respect to chemical characterization of the T. amestolkiae colorants, we are performing experiments that are not concluded yet, but give us evidence that the red colorants from T. amestolkiae is similar to the azaphilone colorant N-GABA-PP-V from T. albobiverticillius 30548 (Venkatachalam et al., 2018).
Venkatachalam, M., Zelena, M., Cacciola, F., Ceslova, L., Girard-Valenciennes, E., Clerc, P., Dugo, P., Mondello, L., Fouillaud, M., Rotondo, A., Giuffrida, D., Dufossé, L., 2018. J. Food Comp. Anal. 67, 38–47.
- Materials and Methods are well described. The Results are of interest, and report innovative data. Furthermore, data are adequately discussed. However, the major concerns are related to the statistical analyses and to the figures reported in the manuscript.
RESPONSE: We apologize for the missing statistical analyses. We performed measurements in triplicates. We added the standard deviation to the figures reported in the manuscript and averages of the experimental data.
- Figure 1: Please report the standard deviation in the Figure. If there are graphical problems, consider to insert an additional table in which the raw measurements are reported with the correspective standard deviation. If the authors will decide to insert the supplementary table, they should remember to insert the reference to the new supplementary table in the caption of the figure. Moreover, the authors should perform ANOVA test on the content of the three pigments as a function both of time and stirring speed. Then, they should report different letters on the top of each bar indicating statistical differences. Also in this case, authors should remember to report information regarding statistical analysis in the caption of the figure.
RESPONSE: We apologize by our missing. According to reviewer suggestion, it was included in the revised manuscript all standard deviation and ANOVA analysis.
RESPONSE: We apologize for the missing statistical analyses. We performed measurements in triplicates. In order to insert the error bars, the graphical was changed for columns and the error bars were inserted. The information regarding statistical analysis was inserted in the caption of the Figure.
- Figure 2: Please, report the letter of the panel on the top-right of each panel, and delete the graph “)”. Moreover, since JoF has no additional cost for colour images, consider to redraw the image using different colours for the different treatments. Finally, the authors should perform ANOVA test on the content of the sucrose and on pH values as a function of time. Considering the problems in reporting the letters in each panel of the graph, I suggest to report them in a new supplementary table. Authors should remember to report in the caption of the figure both information regarding statistical analysis and reference to the new supplementary table.
RESPONSE: The reviewer was right. We apologize for the missing statistical analyses. We performed measurements in triplicates. As suggested by the reviewer, a table containing all the results was reported in the Supplementary material. Statistical test Anova one way (with Welch correction for unequal variances) and Tukey post-test were performed. Figure 2 was improved
- Figure 5: Please report the standard deviation in the Figure. Moreover, the authors should perform ANOVA test on the peroxide content as a function of time. Then, place different letters on the top of each bar indicating statistical differences. Also, in this case, authors should remember to report information about statistics in the caption of the figure.
RESPONSE: We thank the reviewer suggest. Statistic date were insert in a Table (Table S2 in the Supplementary material). The error bars were inserted in the Figure.
- Moreover, some minor mistakes were found during the reviewing procedure:
Affiliation section: immediately after the affiliation, the e-mail and the initials of each author must be reported into the brackets (ex. Department of Engineering Bioprocess and Biotechnology, School of Pharmaceutical Sciences, Universidade Estadual Paulista – UNESP, Araraquara, Brazil; fernanda.oliveira@unesp.br (F.O.); caio.azevedolima@unesp.br (C.A.); etc…)
Line 41: “may have harmful effects on human health and in the environment” -> for the environment
Line 43: “cosmetics” -> Authors give information about the use of colorants in food industry, and not in pharmaceutical industry. Please, remove cosmetic, or add information regarding the use of colorants in pharmaceutical industry.
RESPONSE: We apologize for the typing errors and thank the reviewer for the time has taken in consideration of the review. All considerations were taken in account and were made it better written and explained. Also, we removed “cosmetic” from the manuscript.
- Line 43: “Regarding the natural color market, a survey by Grand View Research, Inc. estimates” -> estimated
Line 69: “One of the most interesting characteristics of natural fungal colorants is the potential of their use as a functional food colorant” -> is their potential use as functional food colorant.
Line 70: “they can present antioxidant” -> they may exert
Line 74: “a representative of food contaminant”; Staphylococcus aureus -> Staphylococcus aureus
Line 83-84, 112 and 390 “T. amestolkiae” -> T. amestolkiae
Line 106-108 Report subscripts in the different formulas
Line 128 Talaromyces -> Talaromyces
Line 187. The post hoc test is not Turkey, but Tukey
Section 3: please, rename “Results and Discussion”
Line 194 “T. amestolkiae” -> T. amestolkiae
RESPONSE: We are grateful for the reviewer consideration. According to the reviewer suggestion, all corrections were made, but if some typing error has gone unnoticed, please let us know.
- Line 222-223: “The highest colorant production occurred at 500 rpm of stirring” -> this statement does not have sense since ANOVA analysis was not performed. Please see comment on Figure 1.
RESPONSE: We apologize for the missing statistical analysis. We performed measurements in triplicates. We added the standard deviation to the figures reported in the manuscript and averages of the experimental data. After this analysis, we realized that the results achieved at 500 and 600 rpm was no statistical different. So, the discussion was rewritten.
- Figure 3: please remove the halo from the trend lines. Please, see note to the statistical of the Figure 1.
RESPONSE: We apologize by our mistake. As suggested, this information was placed in the text.
- Line 266: “it is possible to observe the significant increase” -> “it is possible observing a significant increase.
Line 278 I suppose that UA is AU. Check also Table 1
Author contribution: the names of the authors must be reported as acronyms (ex. F.O.)
RESPONSE: We are grateful for reviewer consideration. As suggested by the reviewer, all corrections were made, but if some typing error has gone unnoticed, please let us know.

Reviewer 2 Report
In this manuscript, Oliveira et al. describe the results of the productivity of natural colorants by T. amestolkiae at various stirring speeds in stirred-tank bioreactor. The profiles of the sucrose consumption and the pH sift in the cultures are also discussed. In addition, at 500 rpm of stirring speed, pH was adjusted to 8.0 and the time course of colorant productivity was measured. Furthermore, the cassava starch-based films containing the T. amestolkiae colorants were prepared and characterized. In conclusion, it is stated that the incorporation of the colorants provides oxidative protection in packaged butter. Although there is no contradiction in the assertion in this paper, there are some problems with experimental reproducibility and measurement parameters in the cultures. In addition, there seems to be some errors in the statistical processing of the results. Therefore, it is required several improvements described below for the acceptance for publication of Journal of Fungi.
- In culture of filamentous fungi using bioreactor, it is thought that errors among experiments are likely occur due to the morphology and high viscosity of the culture medium. Are the results shown in Fig. 1 the quantitative results of the amounts of colorants produced in one culture at each stirring speed? It should be shown as an average of at least 3 cultures and the reproducibility should be discussed. A similar problem lies with the results shown in Fig. 3.
- It is difficult to objectively evaluate the state of culture by measuring only the consumption of carbon source and the sift of pH. Biomass and D.O. should be presented, and the viscosity of the culture broth can be measured. At least the biomass at the endpoint should be presented and the relationship with colorant production should be discussed. By doing so, you may be able to consider why the colorant productivity decreased at 600 rpm compared to 500 rpm (It may be due to the stress of hyphal shear and the resulting reduction in biomass).
- There seems to be an error in the statistical analysis applied to the data in Table 2. One-way ANOVA is a method for analyzing the variance of data, Tukey’s test is usually used for multiple comparison among the multiple groups. The data shown in Table 2 is a comparison between the two groups (with or without the colorants). You can examine the significant difference in the mean values between the two groups for the thickness, aw, and total solids, respectively. It should be re-analyzed by an appropriate test method (e.g. Welch’s T-test).
- The resolution of photos shown in Fig. 4 is too low. You should present a clearer phots.
- Does the data presented in Fig. 5 contain the measurement errors? If it is included, you should draw it as an error bar. If not included, it should be re-analyzed and presented.
Minor comments
Throughout this manuscript, there are many formatting errors as described below.
- Names of microorganism should be italic: L74, L84-84, L112, L128, L194, L390, etc.
- Errors about superscripts and subscripts such as molecular formulas: L100, L106, L108, L143, L166, L172, etc.
Errors about abbreviations
- L342-348: The abbreviations aw and aW are mixed. If they have the same meaning, they should be unified.
- You use the abbreviation CFC for cassava starch-based films with colorant additives, CF for ones without colorant additives, CP for conventional plastic, and U for unpacked butter, respectively (L353-354). Therefore, you can use the abbreviations on L369-370 and L377-379 and do not need to repeat those explanation.
- L399: SEM does not appear after that, so it does not need to be an abbreviation.
L172: Is ‘5 x 2 cm2’ wrong? Is ‘5 x 2 cm’ correct?
This manuscript does not include a Discussion section, and the discussions are included in the results section. Therefore, you should rename the section to ‘3. Results and Discussion’(L189). Along with this, the number of the conclusion section must also be changed to 4 (L386).
Author Response
Reviewer #2
- In this manuscript, Oliveira et al. describe the results of the productivity of natural colorants by T. amestolkiaeat various stirring speeds in stirred-tank bioreactor. The profiles of the sucrose consumption and the pH sift in the cultures are also discussed. In addition, at 500 rpm of stirring speed, pH was adjusted to 8.0 and the time course of colorant productivity was measured. Furthermore, the cassava starch-based films containing the T. amestolkiae colorants were prepared and characterized. In conclusion, it is stated that the incorporation of the colorants provides oxidative protection in packaged butter. Although there is no contradiction in the assertion in this paper, there are some problems with experimental reproducibility and measurement parameters in the cultures. In addition, there seems to be some errors in the statistical processing of the results. Therefore, it is required several improvements described below for the acceptance for publication of Journal of Fungi.
RESPONSE: We appreciate the time and all specific points that the reviewer has taken in consideration of the review. We apologize for the missing statistical analysis. We performed measurements in triplicates. We added the standard deviation to the figures reported in the manuscript and averages of the experimental data.
- In culture of filamentous fungi using bioreactor, it is thought that errors among experiments are likely occur due to the morphology and high viscosity of the culture medium. Are the results shown in Fig. 1 the quantitative results of the amounts of colorants produced in one culture at each stirring speed? It should be shown as an average of at least 3 cultures and the reproducibility should be discussed. A similar problem lies with the results shown in Fig. 3.
RESPONSE: We agree with the reviewer that error among the experiments is common in bioreactor. Each batch was performed in duplicate and the measurements in triplicate. The error bars were included in the Figure 1. Figures 1 and 3 were improved.
- It is difficult to objectively evaluate the state of culture by measuring only the consumption of carbon source and the sift of pH. Biomass and D.O. should be presented, and the viscosity of the culture broth can be measured. At least the biomass at the endpoint should be presented and the relationship with colorant production should be discussed. By doing so, you may be able to consider why the colorant productivity decreased at 600 rpm compared to 500 rpm (It may be due to the stress of hyphal shear and the resulting reduction in biomass).
RESPONSE: We agree with the reviewer. Unfortunately, the final biomass was not measured since in this study, fermentative strategies had focused on colorant production and consumption of carbon source, once the fungus tended to grow attached to the metallic surfaces. It was explained in the reviewed manuscript. The information below was also included in the manuscript.
“Generally, in bioreactor cultivation, reproducible kinetics growth of filamentous organisms is difficult to be obtained. This phenomenon occurs frequently due to mycelial aggregation in fermentation broths with long hairy mycelial morphologies that can adhere to surfaces and form a growing biofilm [1]. Many fungal strains grow preferentially on surfaces and will develop thick layers on walls and surfaces in a submerged fermentation process [2]. In the same way, reproducible samples to measure apparent viscosity are difficult to be obtained, hence, it was also deemed appropriate to omit the time course for biomass and viscosity.”
- Vecht‐Lifshitz, S.E.; Magdassi, S.; Braun, S. Pellet formation and cellular aggregation in Streptomyces tendae. Biotechnol. Bioeng. 1990, 35, 890–896, doi:10.1002/bit.260350906.
- Kim, H.J.; Kim, J.H.; Oh, H.J.; Shin, C.S. Morphology control of Monascus cells and scale-up of pigment fermentation. Process Biochem. 2002, 38, 649–655, doi:10.1016/S0032-9592(02)00095-X.
- There seems to be an error in the statistical analysis applied to the data in Table 2. One-way ANOVA is a method for analyzing the variance of data, Tukey’s test is usually used for multiple comparison among the multiple groups. The data shown in Table 2 is a comparison between the two groups (with or without the colorants). You can examine the significant difference in the mean values between the two groups for the thickness, aw, and total solids, respectively. It should be re-analyzed by an appropriate test method (e.g. Welch’s T-test).
RESPONSE: We apologize by our mistake. It was performed a Statistical test Anova one way with Welch correction for unequal variances.
- The resolution of photos shown in Fig. 4 is too low. You should present a clearer phots.
RESPONSE: We are grateful for the reviewer consideration. As, it was not possible present a clearer phots, Fig. 4 was removed to the Supplementary material.
- Does the data presented in Fig. 5 contain the measurement errors? If it is included, you should draw it as an error bar. If not included, it should be re-analyzed and presented.
RESPONSE: We apologize by our mistake. In the manuscript revised version, the error bars were included. Moreover, all the results and the statistic date were insert in a Table (Table S2 in the Supplementary material).
- Minor comments
Throughout this manuscript, there are many formatting errors as described below.
- Names of microorganism should be italic: L74, L84-84, L112, L128, L194, L390, etc.
- Errors about superscripts and subscripts such as molecular formulas: L100, L106, L108, L143, L166, L172, etc.
RESPONSE: We apologize for the typing errors. We appreciate the time and all specific points that the reviewers have taken in consideration of the review. In the manuscript revised version, all the corrections were made.
- Errors about abbreviations
L342-348: The abbreviations aw and aW are mixed. If they have the same meaning, they should be unified.
- You use the abbreviation CFC for cassava starch-based films with colorant additives, CF for ones without colorant additives, CP for conventional plastic, and U for unpacked butter, respectively (L353-354). Therefore, you can use the abbreviations on L369-370 and L377-379 and do not need to repeat those explanation.
- L399: SEM does not appear after that, so it does not need to be an abbreviation.
- L172: Is ‘5 x 2 cm2’ wrong? Is ‘5 x 2 cm’ correct?
This manuscript does not include a Discussion section, and the discussions are included in the results section. Therefore, you should rename the section to ‘3. Results and Discussion’(L189). Along with this, the number of the conclusion section must also be changed to 4 (L386).
RESPONSE: We apologize by our mistake. The manuscript was carefully revised, and the abbreviations were standardized.

Round 2
Reviewer 2 Report
In the revised manuscript, the points I commented have been almost improved, and therefore it is acceptable for publication of Journal of Fungi.